# Synthesis of Mesoporous Silica Using the Sol–Gel Approach: Adjusting Architecture and Composition for Novel Applications

**DOI:** 10.3390/nano14110903

**Published:** 2024-05-21

**Authors:** Yandong Han, Lin Zhang, Wensheng Yang

**Affiliations:** 1Institute of Nanoscience and Engineering, Henan University, Zhengzhou 450000, China; yandonghan@henu.edu.cn (Y.H.); lin.zhang@henu.edu.cn (L.Z.); 2State Key Laboratory of Inorganic Synthesis and Preparative Chemistry, College of Chemistry, Jilin University, Changchun 130012, China

**Keywords:** mesoporous silica, hollow spheres, multilayer spheres, yolk-shell spheres, non-spherical silica, sol–gel approach

## Abstract

The sol–gel chemistry of silica has long been used for manipulating the size, shape, and microstructure of mesoporous silica particles. This manipulation is performed in mild conditions through controlling the hydrolysis and condensation of silicon alkoxide. Compared to amorphous silica particles, the preparation of mesoporous silica, such as MCM-41, using the sol–gel approach offers several unique advantages in the fields of catalysis, medicament, and environment, due to its ordered mesoporous structure, high specific surface area, large pore volume, and easily functionalized surface. In this review, our primary focus is on the latest research related to the manipulation of mesoporous silica architectures using the sol–gel approach. We summarize various structures, including hollow, yolk-shell, multi-shelled hollow, Janus, nanotubular, and 2D membrane structures. Additionally, we survey sol–gel strategies involving the introduction of various functional elements onto the surface of mesoporous silica to enhance its performance. Furthermore, we outline the prospects and challenges associated with mesoporous silica featuring different structures and functions in promising applications, such as high-performance catalysis, biomedicine, wastewater treatment, and CO_2_ capture.

## 1. Introduction

In 1992, Mobile Oil Company Kresge et al. [1] made the pioneering discovery of a highly ordered hexagonal mesoporous silica, denoted MCM-41. This material is characterized by its substantial surface areas, significant pore volumes, and meticulously arranged mesopores, each measuring approximately 4.0 nm. The synthesis of these mesoporous particles involves a modified Stöber method, i.e., the sol–gel approach, specifically through the hydrolysis of tetraethylorthosilicate (TEOS) and the cooperative self-assembly of TEOS derivatives with cetyltrimethylammonium bromide (CTAB) in an alkaline alcohol/water solution. Kresge’s synthetic MCM-41 not only marked a significant advancement in silica materials science, but also inspired numerous scientists to explore the preparation of mesoporous silica with diverse shapes and microstructures. For example, in 1994, Stucky et al. synthesized a series of cage-like mesoporous silicas by combining cationic and anionic surfactants as bi-templates under acidic conditions [2]. In 1998, Zhao et al. employed nonionic surfactants, specifically triblock Pluronic 123 (EO_20_PO_70_EO_20_) copolymers, as templates for the synthesis of SBA-15-type mesoporous silica, with a tunable pore diameter of 5 nm to 15 nm [3]. Subsequently, in 2010, Polshettiwar et al. used TEOS as the silica source, CTAB as the surfactant, 1-pentanol as the co-surfactant, and urea as a catalyst in a mixed solvent comprising cyclohexane and water, which was conducted under microwave-assisted hydrothermal conditions for the synthesis of KCC-1-type fibrous morphology mesoporous silica [4]. It can be seen that the preparation of mesoporous silica through the sol–gel process involves intricate stages, encompassing the complex hydrolysis/condensation of silicon alkoxide, synergistic growth, and the reconstruction of the mesoscopic phase. Consequently, the growth of these mesoporous silica materials proves highly sensitive to various factors such as pH, temperature, aging time, counterions, and the type of surfactant present in the reaction medium [5]. Moreover, although there have been many reports on using nonionic or anionic surfactants as templates for designing and tailoring the size, morphology, and structure of mesoporous silica [6], using CTAB as a template via the sol–gel approach to guide the architecture and functions of mesoporous silica remains relevant and important at present [7]. For example, to exploit novel properties of mesoporous silica in various applications, numerous reports have been presented on the reconstruction of mesoporous silica with intriguing morphologies, such as hollow, yolk-shell, multi-shelled hollow, Janus, and nanotubular. In addition, the reconstruction of the mesoporous silica architecture and immobilization of functional groups onto their surface requires an in-depth understanding of the sol–gel process to achieve novel applications and optimize the performance of mesoporous silica.

In summary, while there have been comprehensive reviews on the synthesis, functionalization, and applications of mesoporous silica, it is imperative to be aware of new findings. In this account, we first summarize the architecture and composition of mesoporous silica materials prepared using the sol–gel approach. Subsequently, we provide an overview of the advancements in mesoporous silica featuring desired functionalities for emerging applications, such as biomedicine, wastewater treatment, catalysis, and CO_2_ capture. Finally, we offer an outlook on the trends and ongoing challenges in the synthesis of mesoporous silica by the sol–gel approach.

## 2. Synthesis of Mesoporous Silica with Different Architectures and Compositions

### 2.1. Synthesis of Hollow Mesoporous Silica Spheres by the Sol–Gel Approach

The sol–gel chemistry of silica has long been focused on manipulating the size, shape, and distribution of silica particles, which is attributed to the simplicity of the operation, mild conditions, and the controlled hydrolysis and condensation of silicon alkoxide, particularly tetraethyl orthosilicate (TEOS). The preparation of mesoporous silica such as MCM-41 involves the presence of CTAB as a structure-directing agent and ammonia as a catalyst to accelerate the hydrolysis and condensation of TEOS. The self-assembly of silanol monomers with the CTAB surfactant results in the formation of colloidal mesoporous silica with diverse architectures. One notable example is the synthesis of hollow mesoporous silica spheres, due to its ordered pore size, spacious cavity, high surface area, and versatile applications in drug storage/release, catalysis, adsorption, and separation, etc. [8,9,10].

Recent studies have revealed that silica particles prepared via the sol–gel process exhibit inherent inhomogeneity. For instance, Haes and co-workers reported a selective etching strategy using hydrofluoric acid (HF) for the preparation of hollow silica spheres [11]. They initially designed organic–inorganic silica particles with a core-shelled structure. The outer layer comprised a pure silica framework from hydrolyzed TEOS, while the sacrificial layer consisted of organic silica from the co-condensation of TEOS and N-[3-(trimethoxysilyl)propyl]ethylenediamine (TSD). The less compact structures of the organic silica frameworks make them susceptible to attack by hydrofluoric acid (HF). Thus, it is evident that the sol–gel approach offers the advantage of creating hollow spheres with a narrow size distribution and stable structures; for example, the combination of the sol–gel approach with the emulsion method, which enhances the stability of the emulsion while simultaneously promoting the hydrolysis and condensation of TEOS during the synthesis of hollow MCM-41 spheres [12,13]. Yang and co-workers reported in 2010 [14] the use of CTAB as a template in ethanol/water solvent to stabilize and direct the hydrolysis and condensation of TEOS oil droplets (see Figure 1a). The diameters and shell thickness of the hollow mesoporous silica spheres (100% SiO_2_) can be tuned by varying the ratio of ethanol to water (Figure 1b–d). Therefore, the sol–gel approach provides effective control over the size, size distribution, and shell thickness of hollow spheres [15,16], which possess the potential for diverse applications, including but not limited to drug delivery systems, catalysts, and photonic crystals.

### 2.2. Synthesis of Yolk-Shell or Multi-Shelled Hollow Silica Spheres by the Sol–Gel Approach

The synthesis of yolk-shell or multi-shelled hollow mesoporous silica spheres has significantly broadened the applications of hollow mesoporous silica across various fields, such as heterogeneous catalysis [17], multifunctional drug delivery [18], high-performance lithium-ion batteries [19], photonics devices [20], and other fields, due to their distinctive attributes, including a large cavity, low density, extensive surface area, excellent permeability, multiple compartments, and the potential for different functionalization from core to shell or shell to shell [21,22]. These complex multilevel architectures can be constructed using different condensed interfaces in the sol–gel method. For instance, Teng et al. reported the preparation of organic–inorganic hybrid mesoporous silica particles using TEOS and bis[3-(triethoxysilyl)propyl]tetrasulfide (BTSE) as precursors, ammonia as a catalyst, and CTAB as a template in an ethanol/water solvent. The mesoporous organic silica shells possess different condensation degrees, owing to the surface exteriors being further condensed in solution under the catalysis of ammonia. Thus, the sacrificial layer, with a relatively low condensation degree, dissolves during incubation in hot water (Figure 2a) or hydrothermal treatment (Figure 2e), resulting in the formation of yolk-shell (Figure 2b–d) [23] or multi-shelled hollow mesoporous organic silica spheres (Figure 2f–h) [24]. Herein, various types of precursors, ratios of ethanol to water, and the incorporation of organic or other functional groups into the sol–gel process containing CTAB play a crucial role. This approach not only facilitates the expansion of complex architectures of hollow mesoporous silica, but also introduces different functional groups into the walls of mesoporous structures, opening up novel and versatile functions in various sophisticated mesoporous configurations. Therefore, the sol–gel approach is superior compared to the soft template method in emulsion, while being relatively simple, but poses challenges in accurately regulating void size, shell thickness, and the distance from shell to shell due to its limited stability in emulsion systems [25].

### 2.3. Synthesis of Non-Spherical Mesoporous Silica by the Sol–Gel Approach

In addition to the abovementioned spherical mesoporous silica (as hollow, yolk-shell, multi-shelled, etc.), researchers have recently explored the design and construction of non-spherical mesoporous silica by the sol–gel approach. For instance, Li et al. reported in 2014 [26] the synthesis of mesoporous Janus structures by utilizing CTAB as a template in the reaction system to initially obtain spherical MCM-41 with hydrophilic groups (-Si-OH). Subsequently, the spherical MCM-41 was used as seeds, and through the continuous dropwise addition of lipophilic BTSE in a single reaction vessel, the lipophilic cubes (organosilica) were grown on the surface of hydrophilic MCM-41 spheres via the sol–gel process (Figure 3a). The obtained mesoporous Janus with “ball@cube” inorganic–organic silica nanostructure not only possesses amphiphilic wettability but also exhibits dual mesoporosity in pores (2.1 vs. 4.5 nm), as shown in Figure 3b,c. In 2018, Zhao and co-workers also successfully synthesized Janus MCM-41 structures comprising hydrophobic mesoporous carbon spheres and hydrophilic mesoporous silica rods using a surface charge-mediated sol–gel strategy (Figure 3d) [27]. In their study, the Janus structures exhibited adjustable rod lengths ranging from 50 to 400 nm, with hydrophobic mesoporous carbon spheres having a maximum diameter of ~150 nm and a pore size of ~10 nm (Figure 3e,f). These findings demonstrated that the obtained Janus silica particles exhibit wettability between hydrophilic and hydrophobic characteristics. The wettability can be precisely tuned by varying the volume ratio of silica rods to carbon spheres in the sol–gel process, making them beneficial for promoting emulsion stabilization due to their surfactant-like role.

In addition, in 2010 Huang et al. [28] found a novel histidine-derived surfactant, N-dodecanoyl-L-histidine (DHis), which exhibited aggregation behavior in aqueous solution. Upon mixing DHis with an appropriate amount of CTAB, the reaction solution gradually transitioned from a transparent homogeneous state to a jelly-like sol, and ultimately evolved to a fibrous gel with a positive charge on its surface (Figure 3g). This sol–gel process facilitated the replication and construction of tubular mesoporous silica nanomaterials (100% SiO_2_) with an inner diameter of approximately 10 nm and an outer diameter of around 30 nm (Figure 3h). The alteration of the reaction temperature resulted in corresponding changes in the inner and outer diameter of the prepared nanotubes (8 to 10 nm and 26 to 30 nm, respectively), while the thickness of the tube walls remained unchanged. This study also demonstrated the universality of this sol–gel replication strategy, as it could be applied to the design and construction of other mesoporous oxide nanotubes such as TiO_2_, CeO_2_, and ZnO.

### 2.4. Synthesis of Supported 2D Mesoporous Silica Membranes by the Sol–Gel Approach

Two-dimensional membranes of mesoporous silica have also garnered significant attention in the field of nanomaterials [29,30,31,32,33,34,35]. In earlier reports, the ordered mesoporous silica membranes were constructed using silicate as a precursor in the air/water [29] or oil/water [30] interfaces, but the resulting mesoporous membranes exhibited pore orientation parallel to the membrane surface due to lower surface energies between sediments and the substrate. However, in terms of the application, such as magnetic, optic, and electric, superior performance is achieved when the porosity is perpendicular to the surface of the membranes [31]. Some research groups have tried to use pyrolyzed graphite [32] or modified glass [33] as the substrate to change the orientation of pores through π-π or hydrophilic–hydrophobic interactions, but the obtained mesoporous membranes typically exhibit a mixed orientation of parallel and vertical pores. A recent breakthrough was achieved by Teng et al. [34], who utilized CTAB as a template and immersed the substrate in a mixture of alcohol/water solvent containing CTAB and TEOS. Under a volume ratio of ethanol/water of 3:5 in the solvent, CTAB micelles could be vertically aligned on the substrate’s surface. Consequently, mesoporous silica membranes (100% SiO_2_) with pores perpendicular to the substrate were formed through ammonia-catalyzed slow hydrolysis/condensation of TEOS (i.e., the sol–gel process) in proximity to the surface of CTAB micelles. Furthermore, based on this principle, the multi-layered mesoporous silica membranes with more intricate structures can also be constructed using the layer–layer assembly sol–gel technique. Moreover, Zhao and co-workers also reported a simple biphase stratification growth strategy for synthesis of mesoporous silica thin 2D membranes [35], in which glass wafer was placed in the aqueous phase as the substrate, as shown in Figure 4a. The pore diameter of the silica thin membranes is very important in practical applications. Here, it can be easily tuned by varying the upper oil phase parameters. For example, when a smaller content of the silica precursor (10% (*v*/*v*) of TEOS/cyclohexane) is used, via slow hydrolysis and condensation, the resultant mesoporous silica membranes have a pore size of ca. 9.4 nm, and vertical mesopore channels with a uniform thickness of ca. 50 nm (Figure 4b). Moreover, these supported 2D mesoporous silica membranes can be easily assembled into a nanofilter device, which shows an excellent performance in separating differently sized nanoparticles or proteins. For example, the small Au nanoparticles with ca. 5 nm or cytochrome c molecules can pass freely through this silica membrane of ca. 9.4 nm, while large Au nanoparticles (ca. 15 nm) or ovalbumin are almost completely blocked [35]. 

### 2.5. Functionalization of Mesoporous Silica by the Sol–Gel Approach

For the development of mesoporous silica in various fields of application, the addition of various functional elements by the sol–gel approach onto the surface of mesoporous silica is often required. For example, co-condensation is one of the most common approaches. In this method, silicon alkoxide with organic groups and TEOS serve as dual silica sources in a mixture of ethanol and water through co-hydrolysis and condensation (i.e., the sol–gel process), and the functional mesoporous silica is formed (Figure 5a). For example, Nooney et al. reported that mercaptopropyltrialkoxysilane molecules can be used as a functional reagent; after co-condensation with TEOS, the functional group (-SH) is introduced onto the surface of mesoporous silica [36]. These prepared functional silica materials with a self-supporting property have high adsorption capacity for mercury and silver. Another example involves using organosilanes containing -CN groups to introduce -CN groups onto the surface of mesoporous silica particles. The subsequent oxidation results in obtaining mesoporous silica with surface carboxylic acids [37]. The silica functionalized with carboxylate groups (-COOH) has a negatively charged surface in neutral to basic environments, which can be used for removing cationic species from aqueous solution. In addition, the -COOH groups in the pore structure also may serve as anchor sites for biomolecules and polypeptide syntheses. The advantage of co-condensation lies in the uniform distribution of functional elements such as -NH_2_, -SH, -COOH, and aliphatic and aromatic groups on the surface of mesoporous silica. However, a disadvantage is observed when silicon alkoxides containing different organic groups coexist with TEOS in alcohol/water solvents. This coexistence may interfere with the hydrolysis/condensation of TEOS due to the repulsion of organic groups against nucleophilic attacks of hydroxyl ions (OH^−^), resulting in low yield and difficulties in regulating particle size, size distribution, and microstructures [38]. 

In addition, the “grafting” strategy of the sol–gel approach is another commonly used method for the functionalization of mesoporous silica, as depicted in Figure 5b. This strategy involves using silicon alkoxides bearing amino, sulfhydryl, or alkyl groups to modify the surface of the prepared mesoporous silica in the growth solution [39]. Various kinds of grafted mesoporous silica were widely used for the removal of metal ions from solution phase. For example, aminopropyl-functionalized mesoporous silica can be used to adsorb Cu ion guests, and mercaptopropyl-functionalized silica can be used for adsorbing Hg ions [40]. In comparison to co-condensation, the grafting method has the advantage of providing better control over the addition process of functional elements on the surface of mesoporous silica by condensing them with the silanol groups of mesoporous silica. However, a drawback is that only limited amounts of functional elements may be added onto the surface of mesoporous silica, due to the restricted number of silanol groups on the silica surface, leading to poor properties and applications. For example, the incorporation of metal into the framework of mesoporous silica is thought to be useful to create catalytic sites, but grafting catalysts sometimes suffer low metal contents, leaching of metal, and difficulty in reaction control of catalytic sites [40]. 

In the co-condensation or grafting methods, functional elements are typically modified on the outer surface of mesoporous silica. However, in certain applications such as adsorption, catalysis, and sensing, it is necessary to accurately modify the functional elements in the inner walls of mesoporous silica to enhance contact with guest molecules. Therefore, the template method for functionalization of mesoporous silica has gained significant attention recently. In this method, the mesoporous template with functional elements acts as a structure-directing agent to dictate the hydrolysis/condensation of TEOS (Figure 5c). As a result, the functional elements localized in the inner walls of the resulting mesoporous silica by the sol–gel process showed better properties. For example, Lu et al. reported [41] that the mesoporous silica film prepared using an oligoethylene glycol-modified diacetylene surfactant as a structure-directing agent exhibits unusual chromatic changes in response to thermal, mechanical, and chemical stimuli. Furthermore, some studies have also demonstrated that confining functional elements in the pores of mesoporous silica not only leads to better functionalization, but also contributes to novel features. For instance, Yang and co-workers reported that mesoporous silica prepared by co-dissolving 9,10-bis(phenylethynyl)anthracene with CTAB molecules in a mixed solvent with different volume ratios of ethanol to water exhibited unique properties; the fluorescent probe not only clearly detected the evolution of CTAB micelle phases in the mesopores, but also established a correlation between the crystalline state of CTAB (crystalline, glassy, and amorphous) in mesoporous particles and the color rendering of the fluorescent probe [42]. 

## 3. Applications of Mesoporous Silica

### 3.1. Catalytic Applications

To enhance the catalytic applications of mesoporous silica, various functional groups such as -NH_2_, -SH, and -CN have been modified on the surface of mesoporous silica by sol–gel approaches. Subsequently, noble metals [43], strong acid [44], or other compounds can be loaded into the mesoporous walls, playing a role as heterogeneous catalysts [45,46]. These supported mesoporous catalysts not only improve the dispersion, stability, and catalytic activity in complex reactions, but also serve as channels for the adsorption/desorption of free guest molecules [47].

Recently, Shu et al. reported the loading of nickel clusters on the surface of mesoporous MCM-41 materials, demonstrating favorable catalytic effects [48]. They initially calcined mesoporous silica containing the CTAB template at 300 °C to obtain mesoporous silica with carbon/nitrogen hybridization. Subsequently, nickel ions were loaded into the inner walls at 500 °C and reduced under a hydrogen atmosphere to form Ni clusters in MCM-41 (Ni/NC@MCM), as shown in Figure 6a. Due to the direct functionalization of the templates by the sol–gel approach, the Ni cluster (approximately ca. 2.4 nm) as a functional element not only dispersed into the inner wall of mesoporous MCM-41, but also enhanced the adsorption and activation of H_2_, resulted in a 100% hydrodearomatization conversion and nearly 100% hydrogenation yield at 100 °C, as illustrated in Figure 6b,c.

In addition, the green catalytic reaction in the energy field has also become a popular research topic [49,50]. For example, Guo et al. recently reported an effective conversion of glucose to 5-hydroxymethylfurfural (HMF) by grafting a double acid catalyst using the sol–gel approach onto the surface of mesoporous silica [51]. They initially grafted Ti^4+^ ions onto the surface of mesoporous silica to form Lewis acid sites. Subsequently, they phosphorylated these particles, resulting in the formation of a titanium phosphate layer acting as a Brønsted acid site, with the chemical formula Ti_2_O_3_ (H_2_PO_4_)_2_·2H_2_O, as illustrated in Figure 7. In the catalytic process, glucose is initially converted to fructose using Lewis acid as the catalyst. Subsequently, fructose can be further dehydrated to form HMF using Brønsted acid as the catalyst. Compared to conventional solid acid catalysts, this novel titanium phosphate-modified supported catalyst with two different acid sites achieves a high yield of 71% for 5-hydroxyformylfurfural. Additionally, this double acid catalyst prepared by the sol–gel grafting method exhibits good hydrothermal stability, with no significant porous collapse or performance loss observed in over three consecutive runs.

### 3.2. Applications in Nanomedicines

In addition, mesoporous silica nanoparticles prepared by the sol–gel approach are considered highly promising drug carriers due to their excellent biocompatibility and activity, and low susceptibility to immune responses. These mesoporous silica nanoparticles offer tunable pore size, high surface area, large pore volume, and easy surface functionalization for effective drug delivery [52]. In the loading of anti-tumor drugs, the -NH_2_, -COOH, -SH, or other groups are initially modified on the surface of mesoporous silica by sol–gel methods. Subsequently, additional functional elements, such as Ca^2+^, -S-S-, and PEG, are introduced onto the surface of mesoporous silica to establish a delivery response to pH, GSH, temperature, etc. (see Table 1) [53,54,55,56,57,58,59,60,61,62,63]. In recent years, enhancing therapeutic effects has led to a focus on constructing multifunctional drug delivery platforms. For example, Gowsalya et al. reported [64] the incorporation of paclitaxel (PTX) and indocyanine green (ICG) into mesoporous silica by the sol–gel approach, creating a novel multifunctional anti-cancer agent for cervical cancer receptors (Figure 8a). This prepared anti-cancer agent not only facilitates targeting of the nucleus, but also exhibits a compound therapeutic effect through multiple modules, such as photodynamic/chemotherapy/immunotherapy, as illustrated in Figure 8b. The combination of these modules with paclitaxel, an anti-cancer drug, produces a positive therapeutic effect. Therefore, the construction of such a multifunctional drug delivery system by the sol–gel approach not only addresses the limitations of mono-therapy, but also offers new hope for the complete cure of cancer. 

Moreover, recent studies have revealed that cancer stem cells (CSCs), particularly small groups of CSCs within tumor tissues, play a crucial role in promoting anti-tumor resistance [65]. To address these challenges, Chen et al. designed a bio-responsive mesoporous silica loaded with Dox and shABCG2 genes by the sol–gel approach, targeting CSCs. This design effectively kills and eliminates tumor cells both in vitro and in vivo [66], as shown in Figure 9a. The design concept involves initially modifying the mesoporous silica to encapsulate Dox within the pores using disulfide bonds. Subsequently, the Dox-loaded mesoporous silica is further modified with a low-molecular-weight cationic polymer, such as polyethyleneimine (PEI), to enhance the binding capability of the shABCG2 gene to mesoporous silica. Given the high GSH content in tumor cells, this leads to the cleavage of disulfide bonds, facilitating the internalization and release of Dox and shABCG2 in the cytoplasm, as shown in Figure 9b. Therefore, the continuous release of the shABCG2 gene not only blocks the excretion of Dox in the cytoplasm, thus maintaining a higher concentration of Dox in stem cells, but also destroys the self-renewal ability of CSCs, making it difficult for CSC tumors in vivo to recur. 

### 3.3. Environmental Applications

It is well known that environmental issues are increasingly impacting human life and health [67]. In particular, the rise in carbon emissions leading to the greenhouse effect poses a direct threat to human survival. Consequently, research on CO_2_ capture is imperative [68,69]. Among the various methods employed for CO_2_ capture, the adsorption method has garnered significant attention [70]. For example, using mesoporous silica as an adsorbent offers many advantages, due to its ordered pore structure, high surface area, large volume, and thermal stability [71]. Nevertheless, the interaction between the surface of mesoporous silica and CO_2_ molecules is notably weak, which is primarily attributed to physical adsorption. Consequently, the pivotal challenge lies in constructing stable and cost-effective adsorbents through the functional mesoporous silica using the sol–gel approach. It has been found that basic amino compounds can be successfully grafted onto the surface of mesoporous silica by the sol–gel approach [72]. For example, Oliveira et al. reported [73] the grafting of chitosan onto the surface of mesoporous silica by the sol–gel method, followed by the carbonization of chitosan, which endowed the materials with a notably high and stable CO_2_ absorption rate. For instance, when mesoporous silica was impregnated with 50 wt% chitosan at room temperature, the CO_2_ absorption reached approximately 0.78 mmol/g. Thus, employing mesoporous silica with diverse nanostructures as carriers for immobilizing chitosan and deriving adsorbents has proven not only cost-effective, but also environmentally sustainable, making it an increasingly attractive option for CO_2_ capture. 

Another crucial environmental concern is the degradation of petroleum-based polymers, such as plastics. Generally, plastics pose challenges for degradation due to their inherent durability and chemically inert surfaces [74]; this is particularly true for microplastics, which are characterized by sizes smaller than 5 nm, rendering them more harmful [75]. For example, millions of tons of microplastic are generated every year in water ecosystems [76]. Therefore, addressing the management of microplastic waste in wastewater becomes pivotal in mitigating the proliferation of microplastic pollution. Recent studies have demonstrated the efficacy of plastic-degrading enzymes, such as polyethylene terephthalate (PET) depolymerases, in efficiently breaking down microplastics [77,78]. However, the widespread application of free PET enzymes for large-scale PET degradation faces challenges due to issues related to reusability and catalytic stability. Therefore, there is a requirement to immobilize plastic-degrading enzymes on the surfaces of nanomaterials [79], especially on the surface of mesoporous silica by the sol–gel method [80]. Goddard et al. reported [81] the direct immobilization of the enzyme on the surface of mesoporous silica by grafting the polyethylene terephthalate (PET) enzyme binding with a peptide (SBP), as shown in Figure 10a. Furthermore, the introduction of PET enzyme-fixed mesoporous silica into the lining material of wastewater treatment tanks was found to enhance the degradation of microplastic waste (Figure 10b). These reports demonstrated that the mesoporous silica, which was utilized as a carrier and subjected to surface functionalization by the sol–gel method, was effective in addressing environmental challenges and contributing to improvements in overall health levels. 

### 3.4. Other Applications

In other applications of mesoporous silica, the manipulation of electronic devices, particularly within the domain of flexible electronic devices, has emerged as a popular research topic [82,83]. Generally, flexible electronic devices include deformable conductive polymers, liquid metal-embedded elastomers, and self-healing hydrogels [84,85]. Among these, self-healing hydrogels are recognized as the most suitable candidates for flexible electronic devices owing to their exceptional ionic conductivity [86]. However, conventional hydrogels exhibit low toughness, with breaking energies less than 10 J/m^2^ [87]. Hence, the imperative research focus lies in the development of hydrogels with enhanced resilience and self-healing capabilities. Recently, Lin et al. reported [88] the preparation of stretchable conductive hydrogels with both high strength and superior performance. This was achieved by incorporating carbon nanotubes (CNTs) into the elastic matrix, resulting in high conductivity approaching 1000 S/m and an impressive stretchability of up to 200%. In addition, Liu et al. also reported a new type of hydrogel [89] by embedding lemon essential oil (EO) within the pores of mesoporous silica (MS) (Figure 11a,b), resulting in the formation of an essential oil-loaded mesoporous silica complex (EO@AMS). Subsequently, this complex was mixed with multibranched polyacrylate and aluminum ions, ultimately leading to the creation of EO@AMS/polyacrylate hybrid hydrogel materials (Figure 11c,d). These mesoporous silica hydrogel materials created via the sol–gel process exhibit notable mechanical strength, conductivity, self-healing capability, and antibacterial properties, as shown in Figure 11e. It can be seen that the incorporation of lemon essential oil into mesoporous silica not only improves the tensile performance of polyacrylate hydrogel, but also extends the duration of its antibacterial effectiveness.

Finally, an interesting research focus in recent years involves the design and development of mesoporous silica with polyacid functionalization. Serving as a highly efficient proton conductor, these mesoporous silica materials demonstrate distinctive properties in various domains, including catalysis [90], selective adsorption [91], and controlled drug release [92]. The functionalization of polyacids is achieved through the sol–gel approach [93,94]. For example, Richard et al. reported [95] the utilization of a robust polyacid with a precisely controlled length—namely, polystyrene sulfonic acid (PSS)—as the raw polyacid material, while they employed a complex consisting of polyacid-based dihydrophilic block copolymer (DHBC) and an oppositely charged polyelectrolyte as a co-template via the sol–gel process for the preparation of polyacid-functionalized mesoporous silica. The study revealed the successful formation of strong polyacid mesoporous silica (PIC/PSS). Following the removal of the template, the microstructure of the pores remained intact (Figure 12a–d) and the acid properties within the pores were preserved, as shown in Figure 12e. The acid density in the mesoporous silica was exceptionally high, equating to one -SO_3_H group per cubic nanometer volume. It demonstrated super-proton conductivity at 363 K and 95% relative humidity, and the conductivity reached an impressive value, i.e., σ = 2.4 × 10^−2^ S·cm^−1^, maintaining consistent performance over a duration of 7 days. Moreover, the pore properties of this acid-functionalized mesoporous silica can be easily adjusted by altering the pH, ion concentration, or strength in the sol–gel medium. Therefore, the polyacid-functionalized mesoporous silica prepared through the sol–gel approach holds promise for paving a new pathway in the design and preparation of future lithium-ion or sodium-ion conducting materials.

## 4. Conclusions and Outlook

Mesoporous silica, with tailored sizes and pore diameters, varied compositions, and diverse architectures, has garnered considerable attention in recent research. In this review, we comprehensively outlined the synthetic pathways of the sol–gel approach, the underlying mechanisms of hydrolysis and condensation of TEOS, and the key considerations for the preparation of single and complex mesoporous silica architectures, including hollow, yolk-shell, multi-shelled hollow, Janus, nanotubular, and 2D membrane. The design principles are primarily based on the sol–gel approach, involving the hydrolysis, condensation, and co-operative self-assembly of TEOS derivatives with CTAB, and the reconstruction of mesoscopic phases. These mesoporous architectures can be successfully obtained by thoroughly controlling the synthetic conditions, such as pH, temperature, aging time, counterions, and type of surfactant in alkaline alcohol/water solution. Moreover, the application potential of mesoporous silica with the desired functionality using the sol–gel method, particularly in catalysis, drug delivery, wastewater treatment, and CO_2_ capture, were also discussed. 

Despite significant progress achieved in both the synthesis and applications of mesoporous silica, several challenges still persist. For example, in catalytic applications, mesoporous silica prepared by the sol–gel approach exhibits relatively limited thermal and chemical stability. Although efforts to enhance stability have been made through the synthesis of mesoporous silica under hydrothermal or solvothermal conditions, large-scale production remains challenging. Therefore, a deeper understanding of the sol–gel approach in reactions could allow the utilization of highly active precursors for the synthesis of mesoporous silica, which could represent a feasible approach. In addition, the accessibility of active sites and the diffusion of reactant and product molecules play pivotal roles in catalysis. Specifically, if mesoporous silica with a 3D accessible surface is employed, it can be utilized for preparing single-atom molecular catalysts. Although applications such as stereoselective and green catalytic reaction have been mentioned, their full potential in mesoporous silica should be further explored and investigated. 

In biomedical applications, successful drug delivery relies on the well-controlled preparation of mesoporous silica by the sol–gel approach. It is crucial to synthesize well-defined structures and drug-loaded counterparts with customizable size, architecture, pore geometry, and surface properties. The synthetic method should be simple, stable, and scalable to meet potential future industrial demands. Additionally, establishing host–guest interactions of mesoporous silica with therapeutic agents is crucial for achieving high drug loading and a rational release profile. Furthermore, before implementation in clinical practice, extensive preclinical studies are required, including comprehensive chemo/physical characterizations of the synthesized mesoporous silica and its effects on diagnosis, treatment, and biocompatibility. It is essential to conduct thorough research on acute/chronic toxicity, as well as changes in genotoxicity resulting from the systemic administration of mesoporous silica, aspects that should not be overlooked.

In environmental applications, amine-functionalized mesoporous silica via the sol–gel approach exhibits considerable promise for CO_2_ separation and capture, attributed to the strong adsorption properties of surface amine groups and the well-ordered mesopore structure. However, the synthesis and characterization of amine-functionalized mesoporous silica are intricate processes, and much remains unknown regarding amine loading levels, pore structures, gas permeation mechanisms, and their kinetics. Based on the results obtained to date, it is believed that amine-functionalized mesoporous silica, in conjunction with polymers, zeolites, metal–organic frameworks, and mixed-matrix materials, represents a technologically scalable platform. 

In summary, compared to other mesoporous oxide materials, mesoporous silica prepared through the sol–gel approach using CTAB as a template offers greater ease in diversification and functionalization. Through meticulous regulation of the hydrolysis and condensation of TEOS in ethanol/water solvent, i.e., the sol–gel process, the design and construction of mesoporous silica with controlled architecture and composition, along with elucidation of a clear structure–activity relationship, promise significant value for novel applications. 

## Figures and Tables

**Figure 1 nanomaterials-14-00903-f001:**
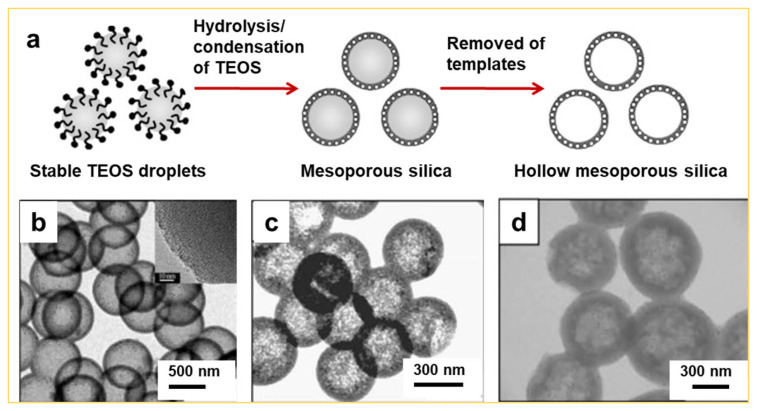
(**a**) Schematic representation illustrating the sol–gel fabrication of hollow mesoporous silica spheres in an emulsion system. (**b**–**d**) TEM images of the hollow silica spheres obtained at ethanol/water volume ratios of (**b**) 0.59, (**c**) 0.53 and (**d**) 0.47, respectively [14]. Copyright 2010, Elsevier.

**Figure 2 nanomaterials-14-00903-f002:**
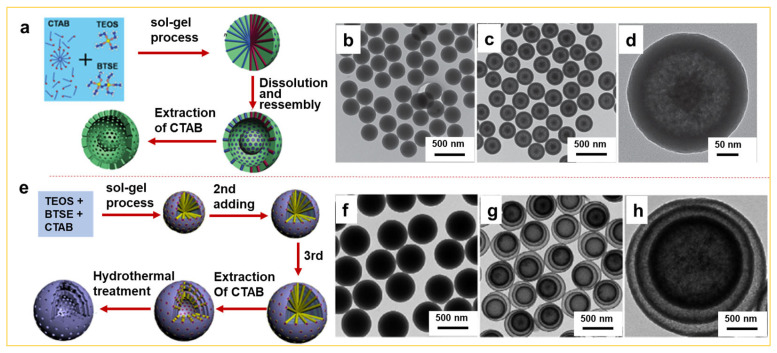
Illustration of the formation process of (**a**) yolk-shell or (**e**) multi-shelled hollow organic silica spheres. TEM image of as-made mesostructured inorganic–organic hybrid silica spheres (**b**), and the yolk–shell mesoporous hollow organic silica spheres (**c**,**d**), which were prepared by incubating the as-made hybrid spheres in water at 70 °C for 12 h [23]. (**f**) TEM image of the mesostructured inorganic–organic hybrid silica spheres synthesized via the sol–gel process by adding BTSE and TEOS three times; (**g**,**h**) the triple-shelled hollow organic silica spheres prepared by hydrothermal treatment for the hybrid spheres [24]. Yolk-shell: Copyright 2014, Elsevier; Multi-shelled hollow: Copyright 2018, American Chemical Society.

**Figure 3 nanomaterials-14-00903-f003:**
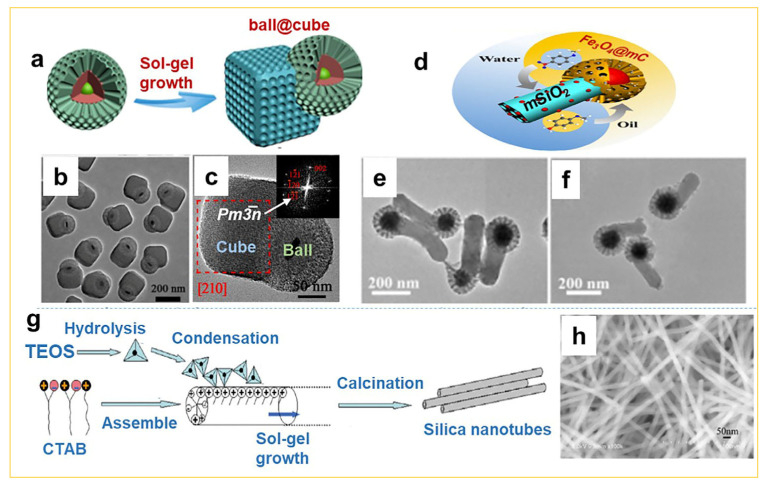
Design and fabrication of non-spherical mesoporous silica architectures. (**a**–**c**) The synthesis of “ball@cube” inorganic–organic Janus silica via the sol–gel approach [26]. (**d**–**f**) The synthesis of Fe_3_O_4_@mC&mSiO_2_ Janus mesoporous silica via the surface-charge-controlled sol–gel encapsulation [27]. (**g**,**h**) The synthesis of tubular mesoporous silica (100% SiO_2_) via the sol–gel template approach [28]. (**a**–**c**) Copyright 2014, American Chemical Society; (**d**–**f**) Copyright 2018, American Chemical Society; (**g**,**h**) Copyright 2010, American Chemical Society.

**Figure 4 nanomaterials-14-00903-f004:**
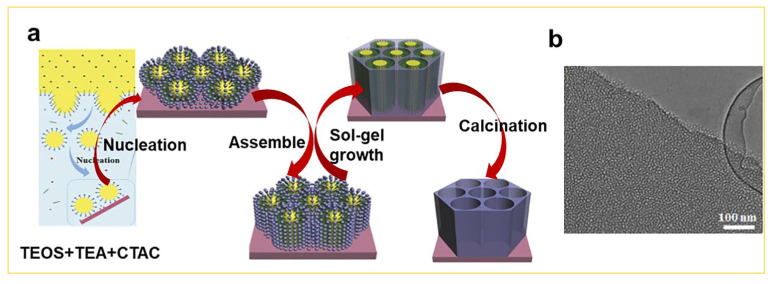
(**a**) The synthesis of mesoporous silica membranous (100% SiO_2_) via the spontaneous sol–gel growth procedure and (**b**) TEM image of 2D mesoporous silica membranes [35]. Copyright 2017, Wiley-VCH.

**Figure 5 nanomaterials-14-00903-f005:**
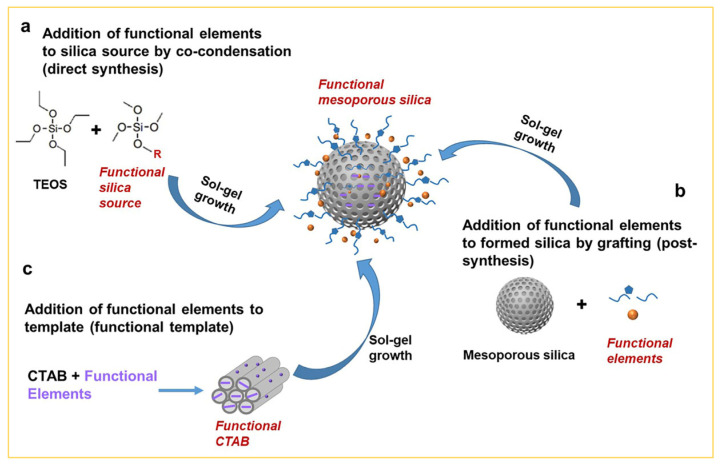
Classification of mesoporous silica MCM-41 functionalization by the sol–gel approach; the functional element can be added at three different stages in the reaction: (**a**) co-condensation, (**b**) grafting, and (**c**) functional template methods, respectively.

**Figure 6 nanomaterials-14-00903-f006:**
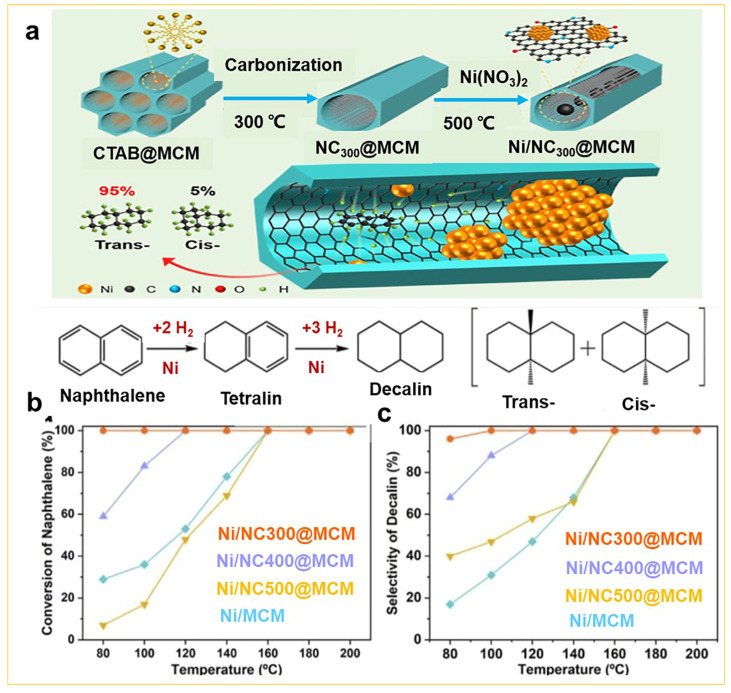
(**a**) The schematic illustration of N-doped carbon interior-modified mesoporous silica with confined nickel nanoclusters for stereoselective hydrogenation. Temperature dependent: (**b**) naphthalene conversion and (**c**) decalin selectivity of different Ni/NC@MCM complexes [48]. Copyright 2022, American Chemical Society.

**Figure 7 nanomaterials-14-00903-f007:**
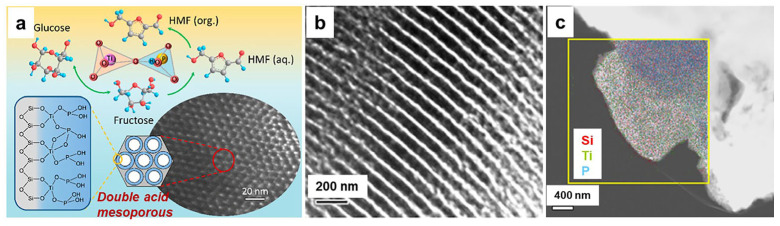
(**a**) The schematic illustration of titanium phosphate grafted onto mesoporous silica as a solid acid catalyst for the synthesis of HMF from glucose. (**b**) TEM image of P–Ti grafted on the mesoporous silica. (**c**) EDX elemental mapping of the P–Ti grafted on the mesoporous silica [51]. Copyright 2022, American Chemical Society.

**Figure 8 nanomaterials-14-00903-f008:**
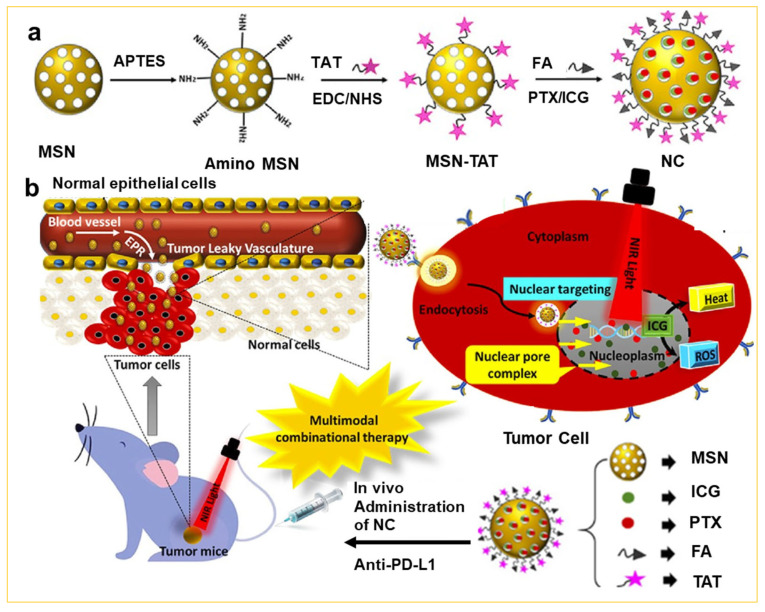
(**a**) The schematic illustration of the synthesis process of near-infrared light-activated and peptide-conjugated mesoporous silica particles (denoted as NC), and (**b**) prepared NC facilitating NIR-mediated multimodal combinatorial therapy [64]. Copyright 2022, American Chemical Society.

**Figure 9 nanomaterials-14-00903-f009:**
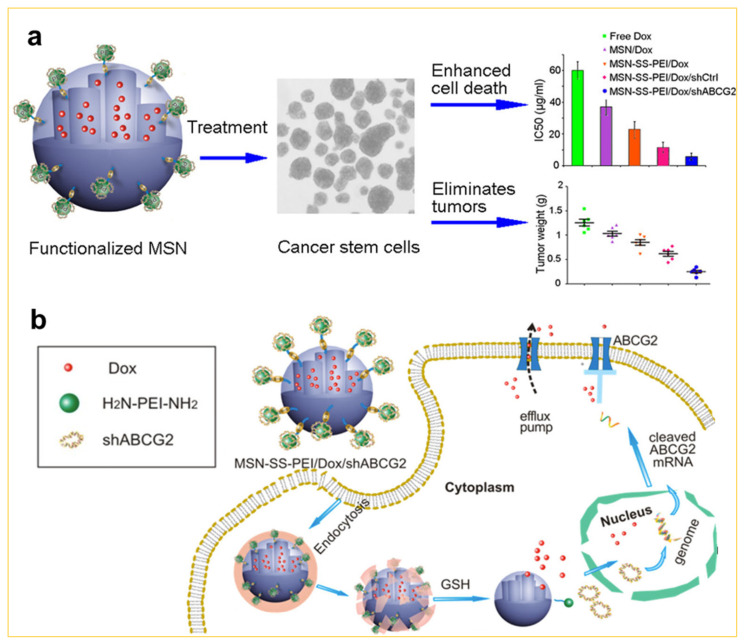
(**a**) The schematic illustration of functionalized mesoporous silica nanoparticles with enhanced sensitivity of cancer stem cells to chemotherapy. (**b**) Several procedures are needed for dual delivery: cell uptake, glutathione-triggered drug release via cleavage of disulfide bonds, and drug diffusing into the cytoplasm and eventually to the nucleus [66]. Copyright 2016, American Chemical Society.

**Figure 10 nanomaterials-14-00903-f010:**
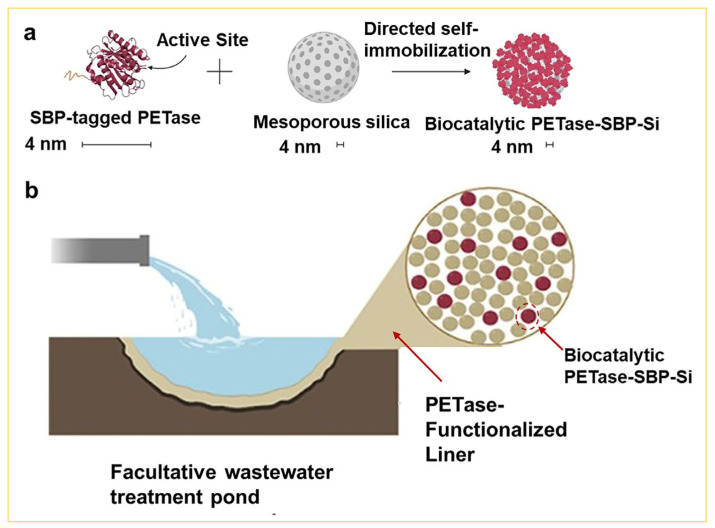
(**a**) The schematic illustration of the immobilization process of SBP-tagged PETase on mesoporous silica. (**b**) Proposed intervention of immobilized PETase for microplastic degradation in wastewater treatment tanks [81]. Copyright 2022, American Chemical Society.

**Figure 11 nanomaterials-14-00903-f011:**
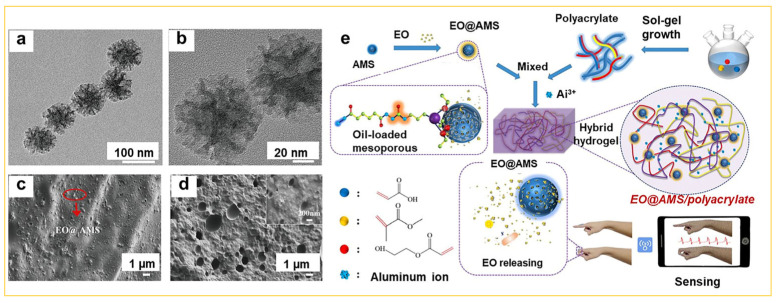
(**a**) TEM image of modification of mesoporous silica with adipic hydrazide (denoted AMS), and (**b**) amplified TEM image of (**a**), (**c**) SEM image of the surface of the EO@AMS/polyacrylate hybrid hydrogel, (**d**) SEM image of the cross-section of the EO@AMS/polyacrylate hybrid hydrogel. (**e**) Schematic illustration of the preparation process of EO@AMS/polyacrylate hybrid hydrogel with self-healing and antibacterial abilities [89]. Copyright 2022, American Chemical Society.

**Figure 12 nanomaterials-14-00903-f012:**
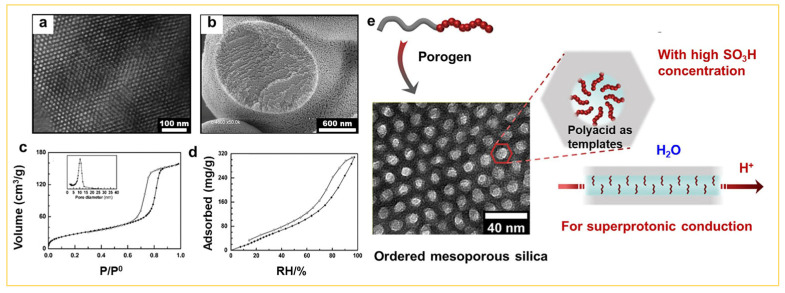
(**a**) TEM image and (**b**) SEM image of the polyacid functionalized mesoporous silica (denoted MesoPIC-PSS). MesoPIC-PSS sorption isotherms of (**c**) nitrogen measured at 77 K and related pore size distribution and (**d**) water measured at 298 K. (**e**) The schematic illustration of the integrated process for structuring and functionalizing mesoporous silica to achieve super-protonic conductivity [95]. Copyright 2022, American Chemical Society.

**Table 1 nanomaterials-14-00903-t001:** Functionalizations of mesoporous silica and their applications in nanomedicines.

Mesoporous Materials	Shape	Diameter (nm)	Pore Size (nm)	Functional Groups	Preparation Method	Medicinal Application	Ref.
Chiral MCM-41	Spheres	100	3.5	-NH_2_/ L-tartaric acid	Co-condensation	Chiral delivery	[53]
MCM-41	Spheres	158	4	-NH_2_/-S-S-/-COOH/ DOX/Ce6	Grafting	Chemo- photodynamic therapy	[54]
MCM-41	Spheres	100	3.4	-COOH/ PEI-AA/ DOX	Grafting	pH-responsive delivery DOX	[55]
MCM-41	Spheres	50	2.5	-NH_2_/ peptides/ DOX	Grafting	Peptides -responsive delivery DOX	[56]
MCM-41	Spheres	184	8.9	-NH_2_/ disulfiram	Co-condensation	Delivery of DSF for cancer therapy	[57]
Hollow mesoporous silica	Hollow spheres	155	-	-NH_2_/GA/PEI/Cur	Grafting	pH-responsive release of curcumin	[58]
Dendrimer- like mesoporous silica	Dendrimer- Like spheres	110	4.0	-SH/ PEG/DOX	Grafting	pH-responsive delivery DOX	[59]
Hollow mesoporous silica	Hollow spheres	170	5.6	-NH_2_/ lactobionic acid/ indocyanine green/DOX	Grafting	pH/NIR dual-responsive release of DOX	[60]
MCM-41	Spheres	140	2.7	-COOH/ Ca^2+^/5-fluorouracil	Grafting	pH-responsive release of 5-fluorouracil	[61]
MCM-41	Spheres	100	2.7	DOX/dopamine	Impregnation	pH-responsive release of DOX	[62]
MCM-41	Spheres	165	3.6	-S-S-S-S-/-NH_2_/ N-acetylgalactosamine	Co- condensation	GSH- responsive release of epirubicin	[63]

## Data Availability

The datasets used during the current study are available from the corresponding author on reasonable request.

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
