# Peer review of "Synthesis of Mesoporous Silica Using the Sol–Gel Approach: Adjusting Architecture and Composition for Novel Applications"

_nanomaterials, 2024, doi:10.3390/nano14110903_

Round 1

Reviewer 1 Report

Comments and Suggestions for Authors

This review reports on material properties obtained by sol-gel approaches. This contribution is interesting ad almost well structured. Nevertheless, before its publication some changes are needed:

·       As a general remark the figure must be enlarged to allow an easier inspection of each of them.

·       Additional information regarding the material composition and purity for each reported material type.

·       Further data regarding the control of material crystalline structure which can be achieved by sol-gel technique for each reported material class.

·       A paragraph regarding the use of sol-gel to grow supported nanomaterials or protective films.

Comments on the Quality of English Language

A general control pertaining to the use English language throughout the paper is also suggested.

Author Response

Dear Reviewer 1,

  Thank you very much for taking the time to review our manuscript, titled " Synthesis of Mesoporous Silica by Sol-Gel Approach: Adjusting Architecture and Composition for Novel Applications" (Manuscript ID: nanomaterials-2995401). We have carefully considered and addressed the points from the reviewers in this revised version (highlighted in blue color). The one-by-one responses to your comments are attached at the end of this letter. We wish this revised manuscript is qualified for possible publication in Nanomaterials.

Thank you very much for your reconsideration!

Sincerely yours,

Wensheng Yang

Institute of Nanoscience and Engineering

Henan University

Zhengzhou 450000

P. R. China

Tel/fax: +86-431-85168185 

Point-by-point response to Comments and Suggestions for Authors

Comments: This review reports on material properties obtained by sol-gel approaches. This contribution is interesting and almost well structured. Nevertheless, before its publication some changes are needed:

Q1: As a general remark the figure must be enlarged to allow an easier inspection of each of them.

Response: Thank you for this kind suggestion. The remark in Figure have been enlarged, as shown in Figure 2, Figure 3, Figure 4, Figure 5, Figure 6, Figure 7, Figure 8, Figure 10, Figure 11 and Figure 12.

Q2: Additional information regarding the material composition and purity for each reported material type.

Response: Thank very much for this helpful suggestion. The material composition and purity for each reported mesoporous silica material have been added in the revised manuscript.

Q3: Further data regarding the control of material crystalline structure which can be achieved by sol-gel technique for each reported material class.

Response: Thank you for this suggestion. In 1992, Kresge et al. reported that used surfactant aggregates as templates to synthesize ordered mesoporous materials, e.g. MCM-41. However, these mesoporous silica materials prepared by the sol-gel technique were completely different from the well-known zeolite materials, which are a family of crystalline microporous aluminosilicate minerals. Although many attempts had been done to obtain zeolite-like mesoporous silica with enlarged pores and ordered structures, but to date, the synthesis of mesoporous silica with crystalline frameworks by sol-gel approach is still a major challenge (see conclusion and outlook in text).

Q4: A paragraph regarding the use of sol-gel to grow supported nanomaterials or protective films.

Response: Thank you for this constructive suggestion. A paragraph regarding the synthesis of supported 2D nanomaterials by sol-gel grow have been added in the revised manuscript.

Response to Comments on the Quality of English Language:A general control pertaining to the use English language throughout the paper is also suggested.

Response: Thank you for this suggestion. We have checked the manuscript carefully and improved the English expressions for better understanding (highlighted in blue color).

Reviewer 2 Report

Comments and Suggestions for Authors

This manuscript is a review in the field of synthesis of mesoporous silica  with unusual architecture for some new applications.

Mesoporous silica is the most studied object of sol-gel chemistry. The authors had a difficult task - to review methods for producing mesoporous silica. Two correct areas were selected: unusual object architecture (hollow spheres, multilayer spheres, egg yolk-type spheres, non-spherical particles), as well as applications of functionalized silica (nanomedicine, environmental).

The manuscript is written in good scientific language, 94 sources were reviewed, mostly from the last 20 years.

The article will be of interest to a wide range of scientist working in the field of sol-gel synthesis and in the field of application of mesoporous silica.

I would like to note the following comments and suggestions:

1. The illustrations given in the article must be enlarged in size; at this scale the information is difficult to read.

2. In my opinion, section 2.4., dedicated to functionalization, could be expanded with additional sources and describe the results obtained in more detail. Since it is precisely such materials that are of significant interest in the areas of application considered by the authors: catalysis, medicine, environmental application.

3. keywords can be made more specific, not as general as the authors give

Comments on the Quality of English Language

Minor editing of English language required

Author Response

Dear Reviewer 2,

  Thank you very much for taking the time to review our manuscript, titled " Synthesis of Mesoporous Silica by Sol-Gel Approach: Adjusting Architecture and Composition for Novel Applications" (Manuscript ID: nanomaterials-2995401). We have carefully considered and addressed the points from the reviewers in this revised version (highlighted in blue color). The one-by-one responses to your comments are attached at the end of this letter. We wish this revised manuscript is qualified for possible publication in Nanomaterials.

Thank you very much for your reconsideration!

Sincerely yours,

Wensheng Yang

Institute of Nanoscience and Engineering

Henan University

Zhengzhou 450000

 P. R. China

Tel/fax: +86-431-85168185 

Point-by-point response to Comments and Suggestions for Authors

Comments: This manuscript is a review in the field of synthesis of mesoporous silica with unusual architecture for some new applications. Mesoporous silica is the most studied object of sol-gel chemistry. The authors had a difficult task - to review methods for producing mesoporous silica. Two correct areas were selected: unusual object architecture (hollow spheres, multilayer spheres, egg yolk-type spheres, non-spherical particles), as well as applications of functionalized silica (nanomedicine, environmental). The manuscript is written in good scientific language, 94 sources were reviewed, mostly from the last 20 years. The article will be of interest to a wide range of scientist working in the field of sol-gel synthesis and in the field of application of mesoporous silica.

Response: We thank Reviewer 2 very much for the encouragement. No action was taken here.

Comments: I would like to note the following comments and suggestions:

This review reports on material properties obtained by sol-gel approaches. This contribution is interesting and almost well structured. Nevertheless, before its publication some changes are needed:

Q1: The illustrations given in the article must be enlarged in size; at this scale the information is difficult to read.

Response: Thank you for this kind suggestion. The illustrations given in the article have been enlarged in size, as shown in Figure 2, Figure 3, Figure 4, Figure 5, Figure 6, Figure 7, Figure 8, Figure 10, Figure 11 and Figure 12.

Q2: In my opinion, section 2.4., dedicated to functionalization, could be expanded with additional sources and describe the results obtained in more detail. Since it is precisely such materials that are of significant interest in the areas of application considered by the authors: catalysis, medicine, environmental application.

Response: Thanks for this constructive suggestion. To better dedicate to functionalization of mesoporous silica by sol-gel approach, we have expanded relevant results obtained in more detail in the revised manuscript (highlighted in blue color).

Q3: keywords can be made more specific, not as general as the authors give.

Response: Thank you for this kind suggestion. The keywords have been replaced with more specific in the revised manuscript.

Response to Comments on the Quality of English Language:Minor editing of English language required.

Response: Thank you for this kind suggestion. We further checked the manuscript carefully and improved the English language for better understanding (highlighted in blue color).
